# Automation for Interpretable Machine Learning Through a Comparison of Loss Functions to Regularisers

## Abstract

To increase the ubiquity of machine learning it needs to be automated. Automation is cost-effective as it allows experts to spend less time tuning the approach, which leads to shorter development times. However, while this automation produces highly accurate architectures, they can be uninterpretable, acting as 'black-boxes' which produce low conventional errors but fail to model the underlying input-output relationships—the ground truth. This paper explores the use of the Fit to Median Error measure in machine learning regression automation, using evolutionary computation in order to improve the approximation of the ground truth. When used alongside conventional error measures it improves interpretability by regularising learnt input-output relationships to the conditional median. It is compared to traditional regularisers to illustrate that the use of the Fit to Median Error produces regression neural networks which model more consistent input-output relationships. The problem considered is ship power prediction using a fuel-saving air lubrication system, which is highly stochastic in nature. The networks optimised for their Fit to Median Error are shown to approximate the ground truth more consistently, without sacrificing conventional Minkowski-r error values.

## 1 Development of Interpretable Machine Learning

Machine learning regression models are increasingly being used in industrial and engineering contexts for high-stakes decision making, automation and control. These methods often produce low conventional error values, yet are known to produce physically inconsistent results which cannot generalise off test set and so cannot be relied upon to model the ground truth of the system. The models are designed to be accurate, not interpretable, and so a human cannot understand how changes in the inputs change the prediction. In real-world applications, where the output can have a direct effect on human life or the environment, model accuracy alone is not sufficient.

Trust is increased if a trained model approximates the true input-output relationships, performing accurately within the bounds of the training data set and beyond it. It has been demonstrated that for many applications minimising traditional error measures cannot guarantee an accurate approximation of the ground truth (Willard et al. 2020). This is due to a poor inductive bias, the inherent prioritisation of one solution over another (Battaglia et al. 2018), produced by conventional error measures which are based on Minkowski-r metrics (Hanson & Burr 1987).

This trust can be increased by manually tuning to remove overfitting or to provide a solution that makes more sense to the user. However, the expert knowledge and domain experience required to properly tune a machine learning method manually are not always available in industry. Genetic algorithms are therefore increasingly used to search a method's hyperparameter space more efficiently

(Yang et al. 2021) (Kumar et al. 2021); which minimise conventional error measures on a test set, often combined with lowering the complexity of the network. This automation exacerbates the lack of interpretability, as models have a large flexibility, and prediction accuracy is prioritised, a low conventional error is achieved without certainty that the method has modelled the correct internal functions. Regularisation hyperparameters can be optimised alongside other neural network parameters (Tani et al. 2021) (Luketina et al. 2016), which increases the search space and creates more flexibility for methods to produce 'accurate' predictions and avoid overfitting.

Common regression regularisation methods are l1 and l2 regularisation and dropout. For l1 and l2 regularisation, large network weights are penalised in the loss function (Nowlan & Hinton 1992). The absolute value of the weights is penalised in l1 regularisation and the squared value in l2, meaning l1 encourages weights towards zero and l2 encourages weights to be small but non-zero. The l1, l2 and elastic net (l1+l2) regularisers improve a networks generality, increasing the applications where the trained methods can be applied, by penalising complexity. Dropout, where a randomly selected subset of weights are optimised at each epoch rather than the full set, improve the generality of the trained models by preventing co-adaption of weight values (Srivastava et al. 2014). Dropout has been shown to be equivalent to l2 regularisation after scaling by Fisher information (Wager et al. 2013), suggesting that the two should not be used in unison. The neural network regularisation methods discussed above aim to improve generality, reducing overfitting by simplifying the relationships modelled by the networks.

Regularisers improve the modelling of the ground truth in scenarios adhering to the assumptions in the proof in Bishop (1995), under which minimum Minkowski-r error values approximate the conditional average of the dataset. This is because the inductive bias from the loss function guides the input-output relationships towards the conditional average, while the regularisation stops overfitting by simplfying the input-output relationships being modelled. However, these assumptions are restrictive and it is noted that few regression applications adhere to them. For example, one assumption is that the dataset is homoscedastic. In scenarios not adhering to these assumptions, network regularisation simplifies the relationships being modelled but this does not necessarily improve the generality, or model the ground truth.

The Fit to Median Error measure (Parkes et al. 2021) produces more interpretable regression, when used in conjunction with conventional error measures. This is achieved by regularising the learnt input-output relationships to the conditional median of the training dataset: the median output value, conditioned on each isolated input variable in turn (Bishop 1995). For many regression applications the conditional medians are a good approximation of the ground truth input-output relationships but as yet it has not been explored as part of an automated approach.

A challenging regression problem is ship power prediction for a vessel using air lubrication to reduce fuel consumption. It is chosen to be used in this study as it violates the assumptions in Bishop (1995), where the noise in the output space is non-Gaussian and heteroscedastic. In this situation, correctly modelling the ground truth and accurate prediction is required but there is limited understanding of that ground truth (Parkes et al. 2018). The literature shows that shaft powering of a vessel can be predicted with average accuracies of between 1.5-5% error with the use of a regression neural network trained with high frequency data from the vessel (Pedersen & Larsen 2009), (Petersen et al. 2012), (Le et al. 2020), (Jeon et al. 2018), (Liang et al. 2019). All neural network applications to ship power prediction in the literature use a combination of local searches and domain knowledge to identify hyperparameter values. The addition of an air lubrication device increases the complexity of the regression problem, as the system interacts with a number of interrelated input variables.

This paper explores the automation of neural network training to a new problem, with a focus on producing a network which accurately models the ground truth. It compares the ground truth representation of a neural network when a genetic algorithm optimises the network's hyperparameters to reduce the Mean Fit to Median Error measure and compares it to standard regularization using l1, l2 and dropout, and to a network optimised to minimise the Maximum Absolute Error. It is illustrated that neural network regularisation methods (l1, l2 and dropout) can be replaced by the use of the Mean Fit to Median performance measure as an objective in the genetic algorithm, reducing the complexity of the search space and producing networks which more consistently model the ground truth.

## 2    Neural Networks Parameters

Previous applications of neural networks to ship power prediction use between 1 and 3 hidden layers (Leifsson et al. 2008) (Parkes et al. 2019), and between 5 and 300 neurons in each hidden layer (Jeon et al. 2018). To provide a sufficiently large search space to allow verification, or otherwise, of these parameters a maximum of 4 hidden layers and 1000 neurons in each layer are used. The majority of the literature treats the problem as time-invariant and use feed-forward networks, so no recurrent parameters are optimised. As the optimiser or activation functions are rarely documented in the literature, the state-of-the-art optimisers and activation functions available in the Keras framework (Chollet et al. 2015) are used in the optimisation, Table 1.

Table 1: Selected Neural Network Hyperparameters

| Hyperparameter | Value or set |
|---|---|
| Layers | [1,4] |
| Neurons in each layer | [1,1000] |
| Epochs | Increasing from 1-20 for increasing generations |
| Early stopping patience | 5 |
| Loss function | Mean Absolute Error |
| Performance measures | Mean Absolute Relative Error, Maximum Absolute Relative Error, Mean Fit to Median Error |
| Optimiser | SGD, Adam (Kingma & Ba 2014), Nadam (Dozat 2016), RMSprop (Hinton et al. 2012), Adagrad (Duchi et al. 2011), Adadelta (Zeiler 2012), Adamax (Kingma & Ba 2014) |
| Activation function | ReLU, sigmoid, softmax, softplus, softsign, tanh, selu, elu |
| l1 & l2 Rates | 0, 0.01,0.001,0.0001,0.00001 |
| Dropout | [0,0.9) |
| Initialiser | Random Normal ($\mu = 0, \sigma = 0.1$) |

The number of epochs and early stopping procedure are not optimised, as there was a need for predictable compute requirements and allowing the optimisation of these parameters leads to unpredictable run times. The number of epochs to train each network increases for increasing generation number in the genetic algorithm, from 1 epoch in the first 15 generations to 20 in the final 15. This was also implemented to reduce compute and it was validated that when more than 20 epochs were allowed, that the early stopping, with a patience of 5, stopped the training within 20 epochs for the majority of networks. The loss function is similarly not optimised, the Mean Absolute Error is used, as the conditional medians are closer to the ground truth input-output relationships in these datasets than the conditional means.

The performance measures, or the genetic algorithm's fitness functions, are the Mean Absolute Relative Error, the Maximum Absolute Relative Error and the Mean Fit to Median Error. Different combinations of these, alongside the use of regularisation parameters in the search space are compared to illustrate the effect of different types of regularisation.

## 3    cMLSGA Parameters

Table 2: Selected cMLSGA Hyperparameters

| Hyperparameter | Value or set |
|---|---|
| Algorithm at Individual Level | HEIA, IBEA |
| Crossover Type & Rate | SBX & DE, 1 |
| Mutation Type & Rate | Polynomial, 0.08 |
| Number of eliminated collectives | 1 |
| Generations between elimination | 10 |
| Population size | 1000 |
| Generations | 300 |
| Proportion elite | 10% |

In this study cMLSGA[1] is selected as it shows the top performance on a range of evolutionary benchmarking problems (Grudniewski & Sobey 2021) and practical problems (Grudniewski & Sobey 2019). Genetic algorithms are increasing used to tune neural network hyperparameters including regularisation parameters for use on new problems (Jin et al. 2004). Many approaches have multiple genetic algorithm objectives, although these all minimise an error measure and a measure of network complexity (Wang et al. 2019) and (Smith & Jin 2014). The use of multiple different performance measures as objectives is yet to be explored in the literature.

Table 3: Genetic Algorithm Approaches

| Approach | Objective(s) | Network Regularisation |
|---|---|---|
| GA**i** | Mean Absolute Error | l1, l2 and dropout |
| GA**ii** | Mean Absolute Error
Maximum Absolute Error | l1, l2 and dropout |
| GA**iii** | Mean Fit to Median Error
Mean Absolute Error | None |
| GA**iv** | Mean Absolute Error
Maximum Absolute Error | None |

Four approaches are investigated in this study, summarised in Table 3, for approach (GA**i**) and (GA**ii**) the genetic algorithm cMLSGA optimises all variables in Table 2, including the l1 and l2 regularisation rate and the dropout rate of the networks. Although it is advised that l2 regularisation and dropout are not used in the same network the genetic algorithms are provided with zero options for all regularisation parameters, to identify if one is preferable in this scenario.

Approach (GA**i**) is a single objective genetic algorithm optimising the Mean Absolute Error which is compared to a multi-objective formulation where the (GA**ii**) approach optimises both Mean Absolute Error and Maximum Absolute Error. For approaches (GA**iii**) and (GA**iv**) no network regularisation parameters are optimised: l1, l2 and dropout rates are all set permanently to zero. They avoid producing networks that have overfitted by the use of two performance metrics as multi-objectives, (GA**iii**) uses the Mean Fit to Median and Mean Absolute Errors to be minimised and (GA**iv**) uses the Maximum Absolute and Mean Absolute. All approaches use 40 CPUs with 2.0 GHz Intel Skylake processors and 192 GB of DDR4 memory, and take less than 3 days, this setup may not be feasible for widespread industrial application, although it is suggested it is within reach of some industries.

# 4   Data

The data used in this study are from a large vessel equipped with the Silverstream® Air Lubrication System. The air lubrication system works through use of fluid sheering to create an air microbubble carpet directly captured within the boundary layer on the ship hull bottom. The bubble carpet reduces the frictional resistance thereby increasing the speed and reducing the shaft power. Compressors provide a constant supply of air to the hull bottom to maintain a uniform bubble carpet operated at the optimal compressor power that maximises the energy balance. The study is performed on both system on and system off datasets, however for brevity only results for system off are presented as they show similar performance.This prediction is required for a baseline determination of how the system is working, but the relationships between the power, weather, ocean and operating conditions are complex and difficult to model.

The variables considered in this study are the shaft power, speed through water, relative wind speed and direction, draught and trim, with shaft power the target variable. These are selected based on a detailed study into variable selection for shaft power prediction (Parkes et al. 2019). The speed through water is selected over the speed over ground, for use as an input variable, as it is more hydrodymanically relevant and its accuracy is validated by comparison to the speed over ground. The dataset is cleaned by removing rows with missing or non-physical values and all datapoints below 0.05 normalised shaft power are removed. The dataset is split into two using the air lubrication system status: system on and system off, where system on is defined as air lubrication system power greater than zero. The system on dataset contains 352,690 datapoints and system off contains 237,962. The data is split into training, testing and validation sets of 70%, 15% and 15% respectively. Each

---

[1]The code for cMLSGA is available at `https://www.bitbucket.org/*******`.

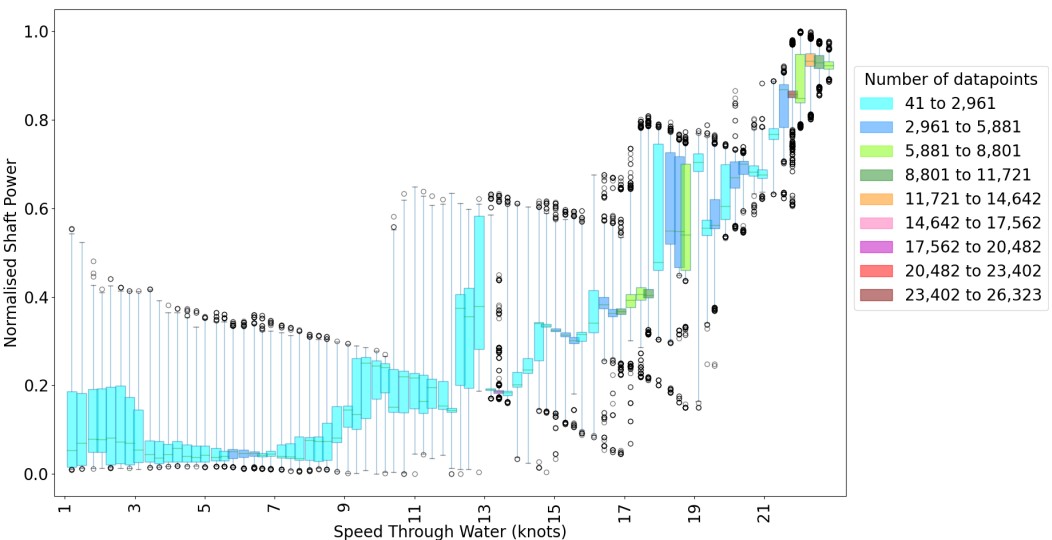

Figure 1: The distribution of the observed shaft powers for half knot bins of speed through the water for dataset where the system is off. In the box and whisker plots the boxes contain 50% of the distribution and the whiskers extend to the datum which is at 1.5 times the interquartile range.

network in the genetic algorithm trains on a randomly sampled 35,000 datapoints from the training set and uses randomly sampled sets of size 7,500 from validation and testing sets for validation during training, and testing to produce the fitness of the network for the genetic algorithm. The errors stated in the paper are from networks on the Pareto fronts of each approach, which are validated on the full testing set.

The datasets contain large regions of sparse data in all input variable domains, this is exemplified by the ship speed domain where each half-knot interval below 16 knots contains less than 0.8% of the data, which accounts for more than half the speed domain, Figure 1. In addition, the boxplot ranges and outliers show high heteroscedicity with idiosyncratic noise caused by situations where the angle of the propeller blades is varied to achieve the required speed. This highlights the complexity in developing models of the powering of this vessel, as the dataset also contains the effects from other latent variables, such as piloting behaviour and route taken.

## 5 Optimisation including regularisation parameters: (GAi) and (GAii)

Previous studies predicting ship powering using neural networks report that l1, l2 and elastic net increase both test set and off-test set errors and that optimal values for both l1 and l2 are zero. Therefore the genetic algorithm setup is biased towards low and zero values of regularisation rates by using a set of exponentially decreasing values and an explicit zero option.

The single objective (GAi) fails to identify that zero regularisation rates produce the lowest errors, favouring networks with the highest possible rate of l2 (0.01), Figure 2b. (GAi) produces networks with the highest Mean Absolute Relative Errors of all the approaches, $(5.19 \pm 0.00)\%$ from Figure 4a. In contrast, (GAii) favours lower l1 and l2 rates of 0 or 0.00001, Figures 2c and 2d, which results in networks with the lowest Mean Absolute Relative Errors of all four approaches, on average, with a value of $(2.87 \pm 0.45)\%$, shown in Figure 4a. This is around 0.5% higher than the lowest documented error for ship power prediction.

It is posited that the high error for the single objective problem is directly related to the use of a large l2 regularisation rate, as noted in previous studies for ship power prediction. It is possible that the use of a multi-objective search algorithm for a single-objective problem means that the optimal hyperparameters can't be found, resulting in large errors. The implementation also requires restrictions in the number of epochs used for training in the initial generations, it is possible this biases (GAi) towards certain size networks, where higher l2 rates are preferable. This hypothesis is

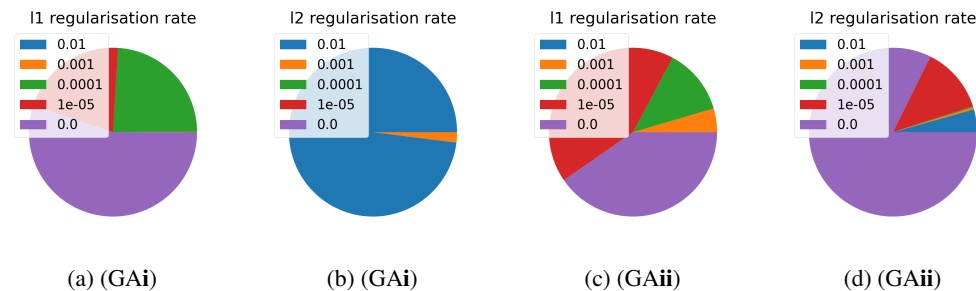

| (a) (GA**i**) | (b) (GA**i**) | (c) (GA**ii**) | (d) (GA**ii**) |

Figure 2: Distribution of regularisation rates for networks in the last 15 generations of (GA**i**) cMLSGA with multi-objectives of minimising Maximum and Mean Absolute Error for (a) l1 and (b) l2 and (GA**ii**) cMLSGA with the single objective of minimising Mean Absolute Error for (c) l1 and (d) l2

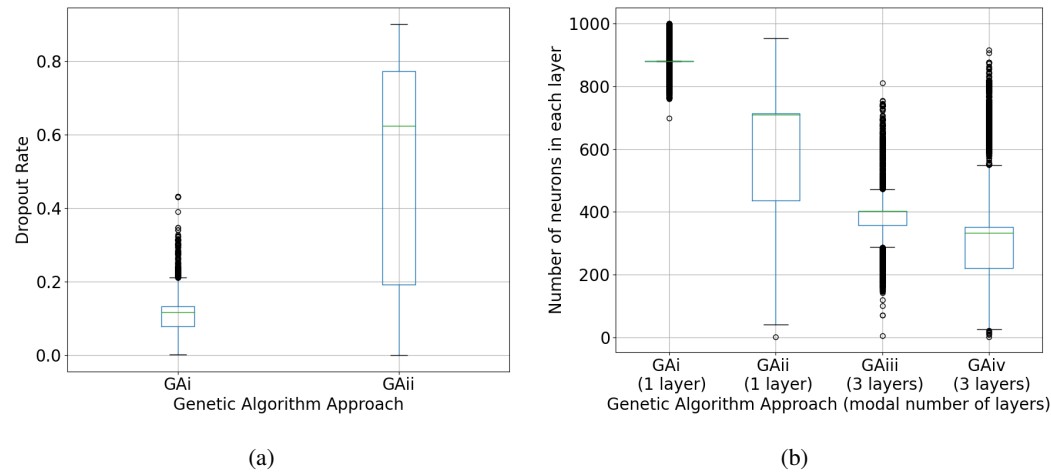

| (a) | (b) |

Figure 3: (a) Dropout rate for networks in the last 15 generations of cMLSGA with (GA**i**) the single objective of minimising Mean Absolute Error and (GA**ii**) multi-objectives of minimising Maximum and Mean Absolute Error and (b) the number of neurons in each layer for networks in the last 15 generations of cMLSGA with (GA**i**), (GA**ii**),(GA**iii**) and (GA**iv**).

supported by the fact that 74.4% of networks in the first 15 generations of (GA**i**) have 1 hidden layer, and that over 99.8% of the networks in the final 15 generations have 1 hidden layer, with $880 \pm 17$ neurons in this layer, Figure 3b. This is significantly more neurons than those in the hidden layer of networks in the final 15 generations of (GA**ii**) which range from 3-952 with a median value of 709, Figure 3b. The added objective of minimising Maximum Absolute Error in (GA**ii**) may cause these slightly smaller networks to be more attractive as they are in a sense regularised by their size, as they have reduced modelling flexibility therefore are less likely to overfit and produce high Maximum Absolute Errors.

Another explanation for the difference in l2 rates chosen by (GA**i**) and (GA**ii**) is the equivalence of l2 and dropout. Since l2 and dropout are equivalent up to a Fisher transformation, their use in conjunction is not recommended. The evidence for this is that (GA**i**) favours the highest l2 rate and has a median dropout rate in the final 15 generations of 0.116, whereas (GA**ii**) favours the zero l2 rate and has a median dropout rate of 0.624, Figure 3a. This illustrates that the genetic algorithms will chose either l2 or dropout to minimise the Mean Absolute Relative Error. The l1 rates also support this hypothesis, as chosen rates for l1 regularisation in the final 15 generations are more comparable for (GA**i**) and (GA**ii**).

# 6 Optimisation using multiple performance measures: (GAiii) and (GAiv)

For approaches (GA**iii**) and (GA**iv**) all neural network regularisation parameters are set to zero. The regularisation is performed by minimising different network performance measures, the Mean Absolute and Mean Fit to Median for (GA**iii**), and the Mean Absolute and Maximum Absolute for (GA**iv**). The trade-off between the two objectives produces regularised neural networks, without explicitly changing the architecture or loss function. The Mean Fit to Median is chosen as it indicates how close the relationships modelled by a network are to the conditional averages of the dataset, in many regression examples this is akin to the ground truth input-output relationships (Parkes et al. 2021). The Maximum Absolute is chosen as for many industrial applications of machine learning the maximum prediction error is more pertinent than the mean error. The Mean Absolute Error is used instead of the Mean Squared Error in both approaches, as the conditional medians are closer to the ground truth input-output relationships in these datasets than the conditional means.

Differently shaped networks are favoured by (GA**iii**) and (GA**iv**), compared to (GA**i**) and (GA**ii**), focusing on networks with 3 hidden layers and on average less than 400 neurons in each layer, Figure 3b. These networks have 51 times the number of connections than the networks chosen in (GA**i**) and (GA**ii**). Apart from (GA**i**), (GA**iii**) has the most consistently sized networks in the final 15 generations, with an interquartile range of 46 neurons, compared to (GA**iv**) which have an interquartile range of 131 neurons. It is suggested that as the Mean Fit to Median Error biases networks towards specific input-output relationships, there is a smaller range of potential network architectures which habitually model these relationships. Whereas networks which minimise the Maximum Absolute Error are less restricted and can model a wider range of input and output relationships.

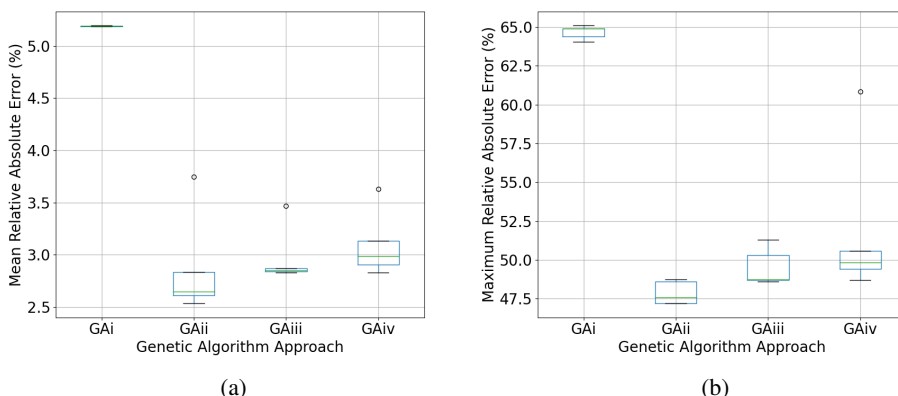

(a)                              (b)

Figure 4: Mean Relative Absolute Error (a) and Maximum Absolute Error (b) from cMLSGA with (GA**i**) the single objective of minimising Mean Absolute Error and (GA**ii**) multi-objectives of minimising Maximum and Mean Absolute Error, both optimising the parameters for l1, l2 regularisation and dropout in the networks, and (GA**iii**) and (GA**iv**) which do not use network regularisation but minimise Mean Fit to Median and Maximum Absolute Error respectively, alongside Mean Absolute Error

The Mean Absolute Relative Errors from networks in the Pareto fronts are $(2.97 \pm 0.25)\%$ for (GA**iii**) and $(3.10 \pm 0.28)\%$ for (GA**iv**). It is expected that (GA**iv**) would produce higher Mean Absolute Relative Errors as discussed above, minimising the Maximum Absolute Error should bias predictions towards the midpoint of the conditional output distributions, whereas minimising the Mean Absolute Error should bias predictions towards the median of these distributions. As it is established that noise in the output distribution is non-Gaussian, Figure 1, these values will not align so some sacrifice in Mean Absolute Error is expected from (GA**iv**). Both (GA**iii**) and (GA**iv**) produce comparable Maximum Absolute Errors, of $(49.5 \pm 1.1)\%$ and $(51.9 \pm 4.5)\%$. It is suggested that this is because, although the conditional median output value and conditional midpoint output value do not align for the majority of the input domain, they are sufficiently close to produce comparable Maximum Absolute Errors.

Across all four approaches, the genetic algorithm producing networks with the highest Mean Absolute Error is the approach which does not provide extra weighting to sparse areas of data. The approaches minimising Maximum Absolute Error are implicitly biased away from networks which predict the majority of the testing datapoints correctly, but predict one datapoint poorly, favouring networks which predict all testing datapoints to a moderate degree of error. Approach (GA**iii**) more explicitly weights prediction in sparse areas of data by favouring networks which model the conditional median of the dataset across all input domains, irrespective of the quantity of data across each input domain. The regression problem of ship power prediction is chosen in part because of it's irregular data distribution; more than 9% of the dataset lies in less than a 0.5 knot interval of ship speed, Figure 1. This provides an explanation for the high testing errors from (GA**i**), where only the Mean Absolute Error is minimised, there is little incentive for the genetic algorithm to produce networks which generalise across the full range of the input domain well.

# 7   Comparison of the interpretability

To asses the interpretability of the networks selected by the four different approaches the learnt relationship between an input, the ship speed, and the output, shaft power, for the networks in the Pareto front of each approach are visualised, Figure 5. These are extracted with the following procedure: set all but one input variable to be constant at the mode; cycle the remaining variable from its minimum to its maximum recorded values with 150 points evenly spaced along the domain and run the new dataset through the trained network.

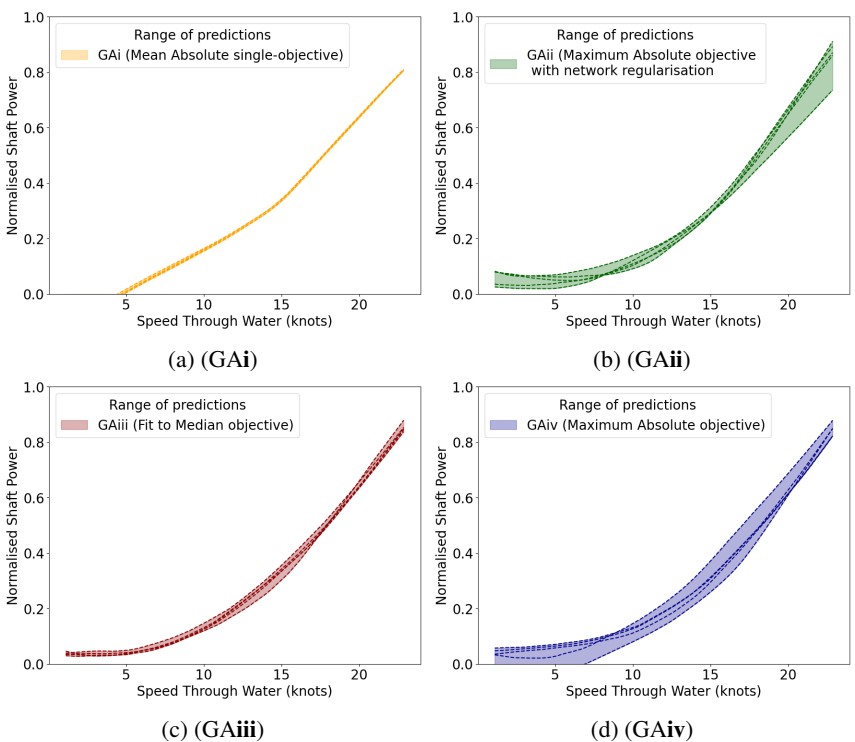

(a) (GA**i**)

(b) (GA**ii**)

(c) (GA**iii**)

(d) (GA**iv**)

Figure 5: The learnt speed-power curves from 5 networks on the Pareto fronts of (GA**ii**), (GA**iii**), (GA**iv**) and the 5 networks producing lowest Mean Absolute Relative Error from (GA**i**).

The approach which produces the most consistent speed-power relationships is (GA**i**), with an average variation of 1.8%[2], Figure 5a. However, the relationship modelled by the 5 networks with the lowest Mean Absolute Relative Error in (GA**i**) all approximate a piece-wise linear relationship which clearly underfits the dataset in Figure 1. The expected trend between ship speed through the water and shaft

---

[2]Average variation in just the speed-power curves are discussed in this section, but it is verified that all input-power curves follow the same trends with variation around 0.5% across input variables.

power is a cubic polynomial, therefore as well as producing the highest Mean Absolute Relative Errors, networks chosen by (GA**i**) model the ground truth input-output relationships the worst out of the four approaches. Both (GA**ii**) and (GA**iv**) produce 5 fairly consistent speed-power curves, with average variations of 5.9% and 10% respectively, Figures 5b and 5d. Both approaches approximate smooth polynomial curves, although the degrees of the polynomials might differ, as multiple curves intersect at various points along the speed axis. The spread of learnt relationships is greater at the highest and lowest speeds for (GA**ii**), with a decrease in spread for speeds of around 15 knots, where many of the curves intersect. The curves from (GA**iv**) show equal spread across the speed domain.

The approach with both accurate and consistent learnt speed-power curves is (GA**iii**), with limited intersections of curves and an average spread of 3.0%. It is suggested that the reason using the Mean Fit to Median Error as an objective in a multi-objective genetic algorithm produces more interpretable results, or more consistent learnt relationships, is because instead of encouraging the networks to model more simple relationships. It encourages the networks to model the conditional median functions of the dataset, supported by the increase in network connections. Whereas the other approaches leave room for networks to fail to model the conditional averages, especially in irregularly distributed and non-normally distributed datasets. The Mean Absolute Error values from networks selected by (GA**iii**) are on average 0.1% higher than those from (GA**ii**), and the Maximum Absolute Error values are 1.6% higher.

A limitation of the approach is that the Fit to Median Error measure will likely perform best at improving interpretability on datasets which violate the assumptions in Bishop (1995); the ship powering example is chosen to illustrate this as it provides a clearly heteroscedastic dataset. For applications where noise profiles are Gaussian, and there is no effect from latent or interrelated input variables, the Fit to Median Error will not improve interpretability, but will perform the same as conventional Minkowski-r metrics, either Mean Squared or Mean Absolute Error depending on the convexity of input-output relationships.

Interestingly, the approach producing the lowest Mean Absolute and Maximum Absolute Errors does not model the ground truth the most accurately. This creates a potential for negative societal impacts, as the standard performance metrics for regression neural networks do not provide a full picture of performance or expected behaviour. Interpretability of trained methods is essential for safe application of machine learning in the real world, especially when automated methods are used to replace experienced professionals. (GA**ii**) demonstrates the same accuracy of approach as those with standard network regularisation, but with a better fit to the ground truth. This approach bypasses the need to use, and therefore to optimise the parameters of the regularisation methods. If evolutionary computation is already being used to optimise network parameters, then compute is saved by removing the network regularisation parameters l1, l2 and dropout. (GA**iii**) completed 300 generations in 46hours whereas (GA**ii**) required 12 hours more computation to complete 300 generations.

## 8 Conclusion

Interpretable and accurate methods are required for widespread application of machine learning to real-world regression problems. To automate the training of neural networks so that the result is interpretable, three different genetic algorithm approaches are compared: one to minimise the Maximum Absolute Error of the networks, which includes standard regularization using l1, l2 and dropout; and two which do not use any network regularisation, one minimising the Mean Fit to Median and one to minimise the Maximum Absolute Error. The results show that all three approaches give similar Mean Absolute Errors from networks on their Pareto fronts, from 2.9% for the approach with regularisation to 3.1% for the approach minimising Maximum Absolute Error. However, the Mean Fit to the Median approach shows a considerably better interpretability, or fit to the ground truth, with a spread in predicted input-output curves of 3% compared to a spread of 6% for the approach using regularisation and 10% when minimising the Maximum Absolute Error.

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
