# OpenReview forum: "Automation for Interpretable Machine Learning Through a Comparison of Loss Functions to Regularisers"
_NeurIPS.cc/2021/Conference — NeurIPS 2021 Submitted_

### Official Review · Reviewer_vAik · 2021-07-15

**Rating:** 3
**Confidence:** 3

**Summary:**

In this paper the authors utilize a new Fit to Median Error measure for a practical machine learning problem of predicting the shaft powering of a vessel using genetic algorithms. They complete an empirical study using different combinations of loss functions and genetic algorithms to demonstrate that the choice of Fit to Median error improves both model accuracy and interpretability.


**Main Review:**

The paper presents a complete empirical study. That said, the reviewer considers the paper lack of novelty and is discussing a very specific applied problem that might not be of the general interests of NeurIPS audience.

The major reasons that lead to the reviewer's rating are as follows.

1. The paper does not define a clear research question.

2. The paper lacks proper related research. Two key concepts covered in the paper are Fit to Median Error and Genetic Algorithms (GAi-Gv) whereas neither of them is novel or clearly elaborated. Therefore it is hard to justify the use of this new error measure, hence the novelty of the paper.

3. The paper does not have enough details to demonstrate how the conclusions or techniques in the paper can be generalized to other machine learning questions. '

====
Post rebuttal:

The reviewer would like to thank the authors for their response, and would like to keep the initial rating.


**Time Spent Reviewing:**

2

---

> ### Author Response · Authors · 2021-08-11
> **Response to reviewer vAik. Main themes: research question, novelty, single application.**
>
>
> 1.The authors define a clear research question in the abstract
>
>     “This paper explores the use of the Fit to Median Error measure in machine learning regression automation, using evolutionary computation in order to improve the approximation of the ground truth.”
>
> This research question is answered as summarised in the conclusion
>
>     “The results show that all three approaches give similar Mean Absolute Errors from networks on their Pareto fronts, from 2.9% for the approach with regularisation to 3.1% for the approach minimising Maximum Absolute Error. However, the Mean Fit to the Median approach shows a considerably better interpretability, or fit to the ground truth, with a spread in predicted input-output curves of 3% compared to a spread of 6% for the approach using regularisation and 10% when minimising the Maximum Absolute Error.”
>
> 2.The authors have performed an extensive review of the literature, finding nothing directly relevant to the current research that is not referenced. Reviewer 5hCU even comments “The paper cites many other works”. A clarification on this comment would be helpful.
>
> 3.The authors chose to focus on one application as the intention of the paper is to illustrate the implications of error measure choices for auto-ML, therefore it was important to discuss the difference in architecture/parameters between the four different approaches. However, previous research [1] bounds the problem types where the mean fit to median approach will be successful and these results should remain representative of that functional space. Outside of this space it is debatable whether these functions remain representative of real-world regression problems.  The authors are willing to apply the methodology to a different application if required and believe the information is available for others to replicate the approach if required. Further explanation on what details are not available would be helpful.
>
> An empirical study is selected as there is no analytical approach which would illustrate the implications of using different error measures in an auto-ML context, due to the stochastic nature of such an approach. Previous research demonstrates the theoretical potential for such a method, but they do not allow a quantitative comparison to current approaches, which can only be shown empirically.
>
> [1] Parkes, A. I., Sobey, A. J. and Hudson, D. A. (2021) “Towards Error Measures which Influence a Learners Inductive Bias to the Ground Truth'” https://arxiv.org/abs/2105.01567

---

### Official Review · Reviewer_5hCU · 2021-07-17

**Rating:** 4
**Confidence:** 3

**Summary:**

The paper claims that building ML models through automation can become a black-box approach that results in lower errors but also lower interpretability. Also, minimizing traditional error measures cannot guarantee an accurate approximation of the ground truth. The paper studies using fit to Median Error measure to produce better model with regard to interpretability of models.

The paper uses ship power prediction problem that aims to reduce fuel consumption, a challenging regression problem, as an application where fit to Median Error measure is used. The paper compares four different genetic algorithms settings: minimizing the Maximum Absolute Error of the networks, minimizing the Maximum Absolute Error and minimizing the Mean Fit to Median.

The experiment results demonstrate that all four settings produced similar Mean Absolute Errors from networks on their Pareto fronts. But the Mean Fit to the Median setting obtained much better interpretability (fit to the ground truth).


**Limitations And Societal Impact:**

Yes

**Main Review:**


The paper is mostly about applying existing techniques to an application. The GA, fully-connected neural network, and different measures are not new ideas. The paper cites many other works.

The paper makes quite many claims. Some of them are quite difficult to comprehend, For instance, in page 8, the paragraph states "Across all four approaches, the genetic algorithm producing networks with the highest Mean Absolute Error is the approach which does not provide extra weighting to sparse areas of data." What is "extra weighting"? The conclusion that GAiii is the approach with both accurate and consistent learnt speed-power curves is a bit stretchy.


It is interesting to read the paper. However, it can be better written. For example, the section title of "3 cMLSGA Parameters" can be "Genetic Algorithm Software cMLSGA Parameter Setting"

The paper uses an application to prove that fit to Median Error measure produces more interpretable model. I found it is not too convincing. The title of paper is about automating interpretable ML through a comparison of loss functions to regularizers. Based on the methodology and experiment results, it is not clear to me that the goal is achieved.



**Time Spent Reviewing:**

5

---

> ### Author Response · Authors · 2021-08-11
> **Response to reviewer 5hCU. Main themes: novelty, written English, use of word 'interpretable'.**
>
> The authors agree that the paper builds on previous research. However, the novelty of the paper is the benchmarking of existing techniques in an Auto-ML context. No studies of this kind exist, with a focus to improve the interpretability (fit to the ground truth), rather than the accuracy, of Auto-ML approaches.
>
> The term ‘extra weighting’ is not defined clearly enough. It is used to explain that the Mean Fit to Median error measure is not a point-based metric. As the Minkowski-r family, for example MAE and MSE, measure the distance between predictions from the training set and the targets, areas of dense data will have more error values within them, so learning methods will be able to model these areas more accurately than the areas of sparse data. The Mean Fit to Median does not measure error in this way, so has equal ‘weighting’, equivalent to an equal quantity of error values across the entire data domain, regardless of the number of datapoints in each region.
>
> The authors are happy to take advice on how to improve the quality of the written communication,  if required.
>
> It is difficult to demonstrate that models are more interpretable, it is a subjective index by nature. The authors believe a key element is the accuracy of the fit to the ground truth, and the probability of fitting to the ground truth, as an understanding of the input-output relationships stops Neural Networks from being a black box. Previous papers indicate that theoretically, and on simulated datasets, that the Mean Fit to Median provides a closer approximation to the ground truth and this is supported by simulations showing that is the case. This has not been quantified or implemented in a practical methodology. The results show a substantial improvement in the repeatability of the new measure, and an indication that this should provide better extrapolation. It indicates that chasing accuracy, as the case for many new methods such as XGBoost, does not help with the interpretability and that networks with a fit to the ground truth will have different architectures but that these networks can be derived from Auto-ML, it does not require user domain knowledge.

---

### Official Review · Reviewer_vvdn · 2021-07-22

**Rating:** 4
**Confidence:** 3

**Summary:**

This paper studies the interpretability of neural networks. Specifically, the authors use four approaches with different objectives and regularizations and use cMLSGA, a genetic optimization method to optimize these problems. Eventually, the authors conduct experiments to validate that Fit to Median Error measure possesses better interpretability compared to L1, L2 and dropout regularizations since it regularizes the learnt input-output relationships to the conditional median.

**Limitations And Societal Impact:**

The authors want to validate that the Fit to Median Error is more interpretable by comparing its performance to other different measures since the authors think that more consistent or accurate models have more interpretability. This logic may not be valid since an interpretability model should tell us why this model makes this prediction. Lipton [1] discussed that interpretability can be achieved if we human can simulate the decision-making process of the model in a limited time. Kim et al. [2] stated that interpretability is the degree to which a human can consistently predict the model’s result. There are also other definitions of interpretability such as interpretability is the degree to which a human can understand the cause of a decision [3]. Theses three are all consistent with what I mentioned, i.e., “an interpretable model should tell us why this model makes this prediction”. For example, a decision tree will tell you why it makes that decision since the final decision is based on the splitting of features on the node of the tree. Merely making the prediction more accurate is not enough for interpretability.

In addition, I think only one dataset may not be sufficient to convince the readers. The authors should also provide more details of the model and method by such as providing formulas and structure of the model.

[1] Z. C. Lipton, “The mythos of model interpretability,” CoRR, vol. abs/1606.03490, 2016.
[2] B. Kim, O. Koyejo, and R. Khanna, “Examples are not enough, learn to criticize! criticism for interpretability,” in NIPS, 2016, pp. 2280–2288.
[3] T. Miller, “Explanation in artificial intelligence: Insights from the social sciences,” Artificial Intelligence, vol. 267, pp. 1–38, 2019.


**Main Review:**

This is a pretty empirical work. The structure of this paper is not conventional but still retains its own logic.

**Time Spent Reviewing:**

4

---

> ### Author Response · Authors · 2021-08-11
> **Response to reviewer vvdn. Main themes: empirical work, use of word 'interpretable', single application.**
>
> An empirical study is selected as there is no analytical approach which would illustrate the implications of using different error measures in an auto-ML context, due to the stochastic nature of such an approach. Previous research demonstrates the theoretical potential for such a method, but they do not allow a quantitative comparison to current approaches, which can only be shown empirically.
>
> The authors agree that interpretability is not a well-defined concept [1], but disagree that “an interpretable model should tell us why this model makes this prediction” as this is an explainable model. The authors have used the term ‘interpretable’ to mean a method which consistently models the same ground truth relationships between inputs and outputs, as we believe this element is essential for truly interpretable machine learning. How can we have interpretable models that remain black-boxes? However, the authors are willing to remove the term ‘interpretability’ from the paper if it is too controversial, and replace the term with ‘consistent’ or ‘reliable’ instead.
>
> It is difficult to demonstrate that models are more interpretable, it is a subjective index by nature. The authors believe a key element is the accuracy of the fit to the ground truth, and the probability of fitting to the ground truth, as an understanding of the input-output relationships stops Neural Networks from being a black box. Previous papers indicate that theoretically, and on simulated datasets, that the Mean Fit to Median provides a closer approximation to the ground truth and this is supported by simulations showing that is the case. This has not been quantified or implemented in a practical methodology. The results show a substantial improvement in the repeatability of the new measure, and an indication that this should provide better extrapolation. It indicates that chasing accuracy, as the case for many new methods such as XGBoost, does not help with the interpretability and that networks with a fit to the ground truth will have different architectures but that these networks can be derived from Auto-ML, it does not require user domain knowledge.
>
>
> The authors chose to focus on one application as the intention of the paper is to illustrate the implications of error measure choices for auto-ML, therefore it was important to discuss the difference in architecture/parameters between the four different approaches. The authors are willing to apply the methodology to a different application if required. However, previous research [1] bounds the problem types where the mean fit to median approach will be successful and these results should remain representative of that functional space. Outside of this space it is debatable whether these functions remain representative of real-world regression problems.
>
> [1] C. Rudin “Stop explaining black box machine learning models for high stakes decisions and use interpretable models instead” Nature Machine Learning, Vol 1, May 2019, 206-215

---

> > ### Comment · Reviewer_vvdn · 2021-08-21
> > **It would be better to change the "interpretability" to "consistent" or "reliable"**
> >
> > As the authors said in their rebuttal, this paper actually does not focus on "interpretability" but rather "more accurate models". Thus I agree that it'd better to change the word "interpretable" into "consistent" or "reliable". Based on the quality of the original submission, I will retain my score.

---

> > > ### Author Response · Authors · 2021-09-09
> > > **The reviewer has defined interpretability incorrectly.**
> > >
> > > The authors specifically state in their rebuttal that the focus of the paper is on interpretability, not accuracy: “The results show a substantial improvement in the repeatability of the new measure, and an indication that this should provide better extrapolation.” The novelty of the paper is in its departure from focussing on accuracy: “It indicates that chasing accuracy, as the case for many new methods such as XGBoost, does not help with the interpretability”.
> > >
> > > The reviewers interpretation that improved fit to the ground truth does not improve interpretability seems to contradict the references they supply. It fits with a number of common definitions found in Lipton [1] but in particular the authors would like to draw the reviewers attention to the following specific examples:
> > >
> > > Miller [3] defines interpretability as “generating decisions in which one of the criteria taken into account during the computation is how well a human could understand the decisions in the given context”. This is the common, black-box, problem which generates much negative press around Machine Learning. An accurate model, such as XGBoost, will map the inputs to the outputs in a way that does not represent the underlying physics and which will not provide smooth changes but where the relationships fluctuate rapidly. This is both due to the separation of the Minkowski-r family metrics from the expected mean when Bishop’s 4 conditions are not met, which is very common in real datasets, and from Jensen’s Inequality. This can’t be interpreted by a human as the relationships do not fit to the ground truth, upon which our understanding of a phenomenon is based, and which blocks us from generating an understanding if it is not already known as the relationships seem arbitrary, because they are (lack of repeatability, for example, shown in the regularization results).
> > >
> > > We propose, in this paper, that it is feasible to automate the machine learning approach to fit to this ground truth, when it is unknown, by changing the loss function, rather than using a point by point accuracy and regularisation. Kim et al. [2] go as far as to say “For instance, fitting models to complex datasets often requires the use of regularization. While the regularization adds bias to the model to improve generalization performance, this same bias may conflict with the distribution of the data.” In removing the regularization, we remove this bias and have demonstrated an improved generalization.
> > >
> > > The authors believe it is a different way to think about Machine Learning with a focus on providing the expected/realistic relationships which are more robust, but that don’t compromise on the point by point accuracy, meaning it can be used in the same manner as current Machine Learning. Indeed Lipton defines the first building block of interpretability as transparency “Informally, transparency is the opposite of opacity or blackbox-ness. It connotes some sense of understanding the mechanism by which the model works.” And this is the key motivation of our research.
> > >
> > > [1] Lipton, The Mythos of Interpretability, https://arxiv.org/pdf/1606.03490.pdf
> > > [2] B. Kim, et al. Examples are not Enough, Learn to Criticize! Criticism for Interpretability, Advances in Neural Information Processing Systems 29 (NIPS 2016)
> > > [3] T. Miller, Explanation in artificial intelligence: Insights from the social sciences, Artificial Intelligence, Volume 267, 2019, Pages 1-38

---

### Decision · Program_Chairs · 2021-09-27

**Decision:**

Reject

**Comment:**

This paper makes a mistake about interpretable machine learning, which is quite often on this topic, in that the interpretation involves many aspects other than accuracy. It requires a major revision of the paper to better reflect the content, thus cannot be accepted in the current form.